# Chemokine Regulation in Temporomandibular Joint Disease: A Comprehensive Review

**DOI:** 10.3390/genes14020408

**Published:** 2023-02-04

**Authors:** Yusen Qiao, Jun Li, Catherine Yuh, Frank Ko, Louis G. Mercuri, Jad Alkhudari, Robin Pourzal, Chun-do Oh

**Affiliations:** 1Department of Orthopedic Surgery, 1st Affiliated Hospital of Soochow University, Suzhou 215005, China; 2Department of Orthopedic Surgery, Rush University Medical Center, Chicago, IL 60612, USA; 3Department of Orthopaedic Surgery, University of Pennsylvania, Philadelphia, PA 19107, USA; 4Department of Bioengineering, University of Illinois Chicago, Chicago, IL 60612, USA

**Keywords:** chemokine, temporomandibular joint disease, inflammation, β-catenin, unilateral anterior crossbite surgery

## Abstract

Temporomandibular joint disorders (TMDs) are conditions that affect the muscles of mastication and joints that connect the mandible to the base of the skull. Although TMJ disorders are associated with symptoms, the causes are not well proven. Chemokines play an important role in the pathogenesis of TMJ disease by promoting chemotaxis inflammatory cells to destroy the joint synovium, cartilage, subchondral bone, and other structures. Therefore, enhancing our understanding of chemokines is critical for developing appropriate treatment of TMJ. In this review, we discuss chemokines including MCP-1, MIP-1α, MIP-3a, RANTES, IL-8, SDF-1, and fractalkine that are known to be involved in TMJ diseases. In addition, we present novel findings that CCL2 is involved in β-catenin-mediated TMJ osteoarthritis (OA) and potential molecular targets for the development of effective therapies. The effects of common inflammatory factors, IL-1β and TNF-α, on chemotaxis are also described. In conclusion, this review aims to provide a theoretical basis for future chemokine-targeted therapies for TMJ OA.

## 1. Introduction 

Temporomandibular joint disorders (TMDs) are conditions related to the masticatory muscles and the temporomandibular joint itself that result in pain and mandibular movement dysfunction [1]. TMDs affect patients’ quality of life by causing chronic muscle pain and limited mouth opening [2]. The prevalence of TMDs, as the second most common musculoskeletal pain, was approximately 31% in adults and 11% in children [3,4]. According to the diagnostic criteria of the Research Diagnostic Criteria for Temporomandibular Disorders (RDC/TMD) and the Diagnostic Criteria for Temporomandibular Disorders (DC/TMD), TMD can be categorized into the three groups: arthrogenous TMD (including disc disfunction and joint pathology), myogenous TMD (masticatory muscle disorders), and headache attributed to TMD [5,6,7]. Temporomandibular joint disorders (TMDs), including disc displacement (DD), and/or osteoarthritis (OA), often lead to joint pain and restrictions in mandibular movement [8,9]. Joint noise or crepitance during jaw movement, as well as less specific signs including ear pain and stuffiness, tinnitus, dizziness, neck pain and headache can also occur [10]. Moreover, oral disease is one of the most common public health issues worldwide, with significant socio-economic impacts [11]. Temporomandibular disorders (TMD) are prevalent and debilitating, and are reported to affect 5–12% of Americans [2], with affliction odds higher for females than males [12], and estimated annual costs of $4 billion [13]. Osseous changes are late-stage characteristics of degenerative joint disease [14,15,16] that are found in the TMJs of 20–30 year olds. This is over two decades earlier than in other human joints [17], and is evidence of the distinct susceptibility of the TMJ. Why TMJ tissues prematurely fail compared to post-cranial joints is not understood. TMJ internal disc derangement (ID) and/or OA, are accompanied by synovitis characterized by chronic inflammatory changes like synovial lining hyperplasia. In these cases, the number of new capillaries and small vessels increases, followed by inflammatory cells infiltrating around these vessels [18,19]. With a frequency of 25%, TMJ OA is one of the most common end-stage TMD pathologies, and the molecular mechanisms associated with TMD remain undiscovered [20]. Chemokines are a family of small cytokines, which play a key role in the exudation of inflammatory cells from the vascular system to tissues at the inflammatory site [21]. More specifically, chemokines can stimulate the chemotaxis of neutrophils, macrophages, and T lymphocytes. In addition, inflammatory cells produce inflammatory cytokines such as IL-1β, which plays a role in OA pathology by promoting the expression of matrix-degrading enzymes and oxidative metabolites, leading to extracellular matrix degradation. Furthermore, inflammatory cytokines can stimulate the synovium and other tissues to produce more chemokines [22,23]. Therefore, this review of the literature aims to provide various chemokines and their role in TMJ OA. Additionally, we also briefly discuss the role of CCL2 in TMJ OA.

## 2. Inflammation and TMJ

TMJ OA is described as low-inflammatory arthritis, while rheumatoid arthritis (RA) is described as a high-inflammatory disease. Inflammation during TMJ OA plays a key role in the onset and progression of the disease and the pain intensity [18]. Although some inflammatory mediators are common in the local TMJ OA and systemic inflammatory diseases, local levels and compositions of these mediators may change. In fact, different cytokines, chemokines, chemokine receptors, enzymes, and bone-stimulating resorption factors that are not reported in RA are considered markers of active TMJ OA [24,25]. Increased hyperemia of capillaries and infiltration of inflammatory cells, such as T cells, monocytes, or macrophages, have been observed in the TMJ synovial lining of patients with TMJ ID and OA. Several cytokines and chemokines were detected in the synovial fluid of patients with ID or OA, which include IL-1β and TNFα [26]. These cytokines have been shown to have a role in the expression of eicosanoid acid, chemokines, and proteins in TMJ synovial cells [27].

## 3. Chemokine and TMJ 

Chemokines are pro-inflammatory peptides (8–14 kDa) that are also known as chemotactic cytokines or chemical hormones [28]. Their main biological function is to recruit leukocytes to local inflammatory sites [29]. All chemokines share major structural similarities, which include a conserved 4-cysteine motif. Some chemokines are proinflammatory cytokines, while other chemokines are thought to maintain self-regulation and control cell migration during normal tissue maintenance and development [30]. According to the arrangement of the conserved cysteine residues of mature proteins, chemokines are divided into four subfamilies: CXC(α), CC(β), C(γ) and CX3C(δ). The corresponding receptors are CXCR, CCR, CR and CX3CR [31]. Among them, α and β family members have the most extensive functions. The CXC subfamily, which has one amino acid(aa) residue separating the first two cysteine residues, includes interleukin-8 (CXCL8), growth associated oncogene GROα (CXCL1), stromal cell-derived factor 1 (SDF-1), and platelet factor 4 (PF-4 or CXCL4). Both IL-8 and GROα can send chemotactic neutrophils to inflammatory regions, while SDF-1 and PF-4 recruit lymphocytes and monocytes. The CC subfamily includes monocyte chemoattractant protein MCP-1 (CCL2), macrophage inflammatory protein MIP-1α (CCL3), MIP-1β (CCL4), exotaxin (CCL11), and RANTES (CCL5). These can be expressed on chemotactic monocytes, basophils, dendritic cells and memory T cells. The third chemokine subfamily, C chemokine, is composed of two members: XCL1 (lymphotactin-α) and XCL2 (lymphotactin-β). The fourth chemokine subfamily (CX3C) is represented by fractalkine [32]. It consists of three amino acids that separate the two cysteine residues. In C chemokine studies, little is known about the structural and functional properties of TMJ disease. 

A chemokine receptor is a transmembrane G-protein-coupled receptor that is selectively expressed on the surface of target cells. In TMJ, the inflammatory cells in synovial tissues of RA, OA and ID are increased. Additionally, some chemokines such as IL-8, GROα, RANTES, and MIP-1 have been detected in human chondrocytes and synovial cells at significantly increased levels. The expression of CXCL8 and MCP-1 have also been reported to be increased in the synovial fluid of patients with TMJ ID and/or OA [33]. Such chemokines can promote the infiltration of inflammatory cells in TMJ joints, which causes the release of degradation enzymes, various oxidative metabolites, and inflammatory cytokines. These can lead to joint structural damage and arthritis. Furthermore, the induction of ELR CXC (CXCL1, 2, 3, 6 and 8) may lead to the recruitment of new, small vessels in inflammatory cells and synovial tissues [30,31].

### 3.1. CXC (α) and CX3C (δ) Subfamilies in TMJ Disease

#### 3.1.1. IL-8

IL-8 is a member of CXC chemokines that activate leukocytes [34,35], which were formerly known as neutrophil activator protein-1 or monocyte derived neutrophil chemokine. It is a chemokine capable of inducing chemotaxis and activating neutrophils, including through direct and trans-endothelial migration, release of storage enzymes, induction of oxygen metabolites, and expression of adhesion molecules [34]. IL-8 can also attract T lymphocytes in vitro [36]. Furthermore, IL-8 is associated with many disease states, predominately angiogenic diseases such as RA. In RA, IL-8 has been shown to cause neutrophil infiltration into synovial fluid and joint inflammation. IL-8 also plays a key role in the pathogenesis of TMJ. Koch et al. reported that, compared with OA patients, the level of IL-8 in synovial fluid of RA patients increased significantly [37]. IL-8 is another type of IL-1β-targeting cytokine, one of the strongest chemokines of neutrophils and T lymphocytes. It can promote monocyte homing and activation in the synovium [36]. IL-8 can cause various pathogenic conditions, such as the release of oxidation products, apoptosis of chondrocytes, the production of MMP-13 by articular chondrocytes, and the loss of proteoglycan and subsequent cartilage degradation [38]. Previous studies have shown that appropriate levels of IL-8 are present in the synovial fluid extracted from patients with internal disorders of the temporomandibular joint. Miho et al. investigated the expression of IL-8 and IL-1 in the synovium of TMJ [39]. The upregulation of IL-6 and IL-8 in SMSCs (synovial mesenchymal stem cells) was also mediated by activation of the NF-κB pathway. Since RA synovial macrophages can produce IL-8,10, Miho et al. conducted exploratory studies to determine whether macrophage-like cells exist in TMJ-inflamed synovial tissues. IL-8 is a powerful neutrophil that attracts and activates cytokines in RA and OA. Chondrocytes secrete a large amount of IL-8 by IL-1 and TNF-α, and IL-8 can promote neutrophil mediated inflammation and cartilage deterioration [39].

#### 3.1.2. SDF-1/CXCR4

Chemokine stromal cell-derived factor-1 (SDF-1) is a small cytokine of the CXC chemokine ligand superfamily. It is mainly expressed by bone marrow stromal cells, which include osteoblasts and endothelial cells [40]. C-X-C chemokine receptor-4 (CXCR4) is a specific receptor of SDF-1. The expression of SDF-1 in synovitis or OA patients increases abnormally. In synovium and articular cartilage tissues, the activation of the SDF-1 and CXCR4 signaling pathways can regulate the expression of various inflammatory factors, which include IL-1, IL-6, TNF-α, and MMPs. IL-1 is involved in joint pathology. The SDF-1 and CXCR4 signaling pathway plays a pro-inflammatory role in the experimental TMJ OA. There may be a potential relationship between the SDF-1-CXCR4 axis and extracellular signal regulated kinase (ERK) signaling pathway. In addition, there is evidence that SDF-1α activates ERK and downstream transcription factors (c-fos and c-jun) through CXCR4, which activates adaptor protein-1 on MMP-13 and leads to cartilage damage in knee arthritis. The expression of SDF-1 was high in the synovium [41,42]. Hosogane and his colleagues reported that SDF-1 may promote cartilage destruction during arthritis through the activation of extracellular, signal-regulated kinase signaling pathway and downstream transcription factors [43]. Chen and his colleagues report that SDF-1 and CXCR4 can promote the development and differentiation of osteoblasts, and induce abnormal changes in subchondral bone in OA [44]. Amd3100 is a non-peptide bicyclic AM, which can specifically antagonize CXCR4 receptor at three major interaction residues around the main ligand binding capsule of CXCR4, which are located at transmembrane domains IV, VI, and VII. Amd3100 can effectively inhibit CXCR4 mediated events, but does not interact with any other chemokine receptors (CXCR1 to CXCR3 or C-C chemokine receptor CCR1 to CCR9). A study by Wei and his colleagues demonstrated that Amd3100 (CXCR4 antagonist of SDF-1 receptor) can alleviate cartilage degeneration [45]. The SDF-1 axis is associated with synovitis in the development of TMJ disease, which could regulate inflammatory factors (IL-1) in the synovium. Blocking the SDF-1 and CXCR4 signaling pathway by Amd3100 could prevent TMD patients from developing synovitis. The expression of SDF-1 was mainly derived from osteoblasts. These results suggest that the increased expression of SDF-1 from osteoblasts promotes osteogenic differentiation. CXCR4 may also cause cartilage degradation. The inhibition of SDF-1 signal by Adm3100 can reduce the osteogenesis of subchondral bone and decrease the expression of MMP13 [42].

#### 3.1.3. FKN (CXCL1)

Fractalkine (CX3CL1) is a chemokine of the CX3C family, and is both inflammatory and nociceptive. CX3CL1 has unique connectivity with CX3CR1, which is a single receptor in microglia. CX3CR1 has been associated with oral and facial inflammatory pain [46]. There is much data indicating that the activation of microglia is an effective method to treat oral and facial pain, and that glial cells play an important regulatory role in orofacial pain signaling pathway [47]. The pathogenesis of RA is a complex process. Proinflammatory cytokines such as Interleukin-1β (IL-1β) and tumor necrosis factor-A (TNF-α) are the central mediators of RA [48]. TNF-α, IL-1β, IL-6 are common cytokines. Chemokine-induced neutrophil, chemokine-1, and keratinocyte derived chemokines (KC) can trigger the release of prostaglandins and sympathetic amines, which directly act on nociceptors and cause excessive nociception [49]. The neurochemokine FKN has a pain-promoting effect in the spinal cord [50,51]. At the same time, mouse models of early-stage collagen-induced arthritis demonstrated central sensitization by releasing IL-1β in the spinal cord. In addition to this, P2X7 receptor and protease play a key role in microglia [52]. The protein levels of FKN and P2X7 receptor in the caudal trigeminal nucleus increased by the persistent inflammatory hypernociceptive model induced by TMJ arthritis. This indicates that in the trigeminal nervous system, the persistent albumin-induced hyperalgesia model in TMJ activated the caudal trigeminal nerve through P2X7, cats, and FKN pathways. The expression of FKN in neurons activates microglia, which induces inflammation and hyperalgesia. This suggests that the promotion mechanism of the nociceptive pathway may involve activation of CX3CR1 receptors in microglia through fractalkine [53,54].

### 3.2. CC(β) Subfamily in TMJ Disease

#### 3.2.1. MCP-1 Chemokines

MCP-1 is a chemokine that is part of the CC subfamily. CC chemokines mainly act on monocytes and lymphocytes [55]. In IL-1β-responsive genes, monocyte chemoattractant protein (MCP)-1 mRNA was induced by IL-1β. It was observed to be highly expressed in stimulated synovial cells. The production of MCP-1 in chondrocytes and synovial cells has been shown to play an important role in joint diseases such as RA and OA through monocyte recruitment [56,57]. It has been suggested that MCP-1 is one of the markers of RA disease activity. The protein production of MCP-1 (also known as CCL2) in TMJ has been studied in vitro and in vivo. Both have shown that MCP-1 protein production increases in a time-dependent manner [58]. This suggests that MCP-1 may accumulate in a conditioned medium of synovial cell cultures in vitro. Furthermore, IL-1β presence was confirmed by injection into the TMJ of rats, which can act as a trigger for inflammation and MCP-1 production in inflammatory joints in vivo [59]. This was also demonstrated after 24 h, where the synovium tissue presented with synovitis-like vasodilation, monocyte infiltration, and synovial intima thickening. Immunohistochemical analysis further showed MCP-1-positive cells in the synovial lining as well as subsets of synovial tissue expressing MCP-1 monocytes or synoviocytes. Overall, MCP-1 is one of the main chemokines for monocyte and macrophage migration to the inflammatory site. Stimulated synovial cells in the TMJ with IL-1β produce and release MCP-1, which is believed to be related to the early stage of TMJ inflammation. Thus, MCP-1 may be the main factors leading to the onset of TMJ chronic pain and synovitis [59].

#### 3.2.2. MIP3α-CCR6

Macrophage inflammatory protein-3α (MIP-3α), also known as CCL20, is a highly upregulated gene for IL-1β and TNF-α [60]. Chemokine receptor 6, also known as CCR6, is a CC chemokine receptor protein encoded in humans by the CCR6 gene. MIP-3α and CCR6 may play a role in the recruitment of monocytes and memory lymphocytes from RA peripheral blood to RA joints, which suggests that expression of the MIP-3α receptor, CCR6, may be associated with RA development [61]. RA synovial tissue contains many leukocytes expressing CCR6 [62,63], and both MIP-3α and CCR6 have been detected in the synovial fluid and synovium of RA patients [64]. The nucleotide sequence of the human MIP-3α promoter region has binding sites for Ets, AP-1, SP-1, and NFκB. This suggests that the expression of MIP-3α was regulated by several signaling molecules [65]. IL-1β, or TNF-α, induces MIP-3α production in human synovial fibroblast-like cells (SFCs) through ERK, p38 MAPK, JNK, and NFκB pathways. Increased levels of MIP-3α may trigger dendritic cells, T cells, and B cells to migrate into the synovial tissue and fluids of TMJ ID patients. This could contribute to the onset and progression of TMJ inflammatory changes. Inflammatory cells produce cytokines, matrix metalloproteinases (MMPs), and reactive oxygen species (ROS) in RA [66,67]. The accumulation of inflammatory cells in synovial tissue may lead to the degradation of this tissue in joints through the production of MMPs and ROS [68]. It has been reported that the migration of CCR6-expressing leukocytes is reduced by approximately 70% after treatment with anti-MIP-3α antibodies both in vivo and in vitro [69]. Therefore, anti-MIP-3α therapy could serve as a new method of interventional therapy for TMJ RA.

#### 3.2.3. RANTES-CCR1 

RANTES is an 8 kDa basic polypeptide in the CC chemokine superfamily. It was originally cloned from antigen-stimulated T cell line, which is also known as CCL5. RANTES can be released by chondrocytes, synovial fibroblasts, and inflammatory cells. RANTES is an effective chemical attractant for monocytes, CD4+/CD45RO+ memory helper T lymphocytes, eosinophils, basophils, and mast cells [70,71]. It has been shown that RANTES is overexpressed in normal adult tissues and increases significantly in inflammatory sites and some tumors [72]. RANTES can promote the migration of monocytes, T lymphocytes, natural killer cells, eosinophils, and macrophages. RANTES can also promote the formation of osteoclasts induced by RANKL. RANTES is also highly expressed in tissues, synovial fluid, and peripheral blood of patients with RA or OA. RANTES can trigger and aggravate the inflammatory immune response by promoting the infiltration of immunocompetent cells, and activating synovial fibroblasts to produce inflammatory mediators. MMP-1, MMP-3, MMP-13, and iNOS released by synovial fibroblasts and chondrocytes promote cartilage degradation [73]. Migration tests confirmed that RANTES was an effective chemokine of RAW264, and was responsible for attracting macrophages to inflammatory sites. C-C chemokine receptor type 1 (CCR1) is the receptor of RANTES, and acts as an inhibitor that impairs the migration of GFP BMSCs into OA cartilage and the rescue effect of GFP BMSC injection. RANTES and CCR1 signaling plays a key and synergistic role in the recruitment of GFP BMSCs into TMJ mouse OA-degraded cartilage [74]. Bone marrow mesenchymal stem cells (BMSCs) have shown to have therapeutic effects on TMJ OA in mice. The recruitment of bone marrow mesenchymal stem cells into cartilage can be attributed to RANTES because the inhibition of CCR1 partially blocks this recruitment, while the presence of CCR1 inhibitors eliminates this recruitment. These findings suggest that RANTES and CCR1 signaling may play a synergistic role in the recruitment of GFP BMSCs to OA cartilage of the TMJ [74,75].

### 3.3. Other Subfamilies in TMJ Disease

#### Chemerin-ChemR23

Chemerins are atypical chemokine ligand agonists that bind to many chemokine ligands. Chemerin acts as a chemoattractant for cells of the myeloid lineage, can signal through a potential adipokine of ChemR23, and can also bind CCRL2 in the absence of signaling [76]. Chemerin and its receptors are ubiquitous in the human body. They play a multi-functional role as chemokines, adipokines, and possible growth factors [77]. It was first isolated from human inflammatory fluid [78], which is involved in angiogenesis, osteoblast formation, and glucose homeostasis [79]. It is also a protein of interest in various medical fields, such as immunology, dermatology, metabolism, and development. Chemerin, a ligand protein of ChemR23, is a G-protein-coupled receptor that is expressed in macrophages (dentate cells), dendritic cells, and natural killer cells (NK) [80]. The interaction between Chemerin and ChemR23 is believed to be closely related to the migration of macrophages and Dendritic Cells (DC) to inflammatory sites, as well as mediating inflammatory signals to articular chondrocytes and endothelial cells (e.g., IL-1β, IL-6, IL-8 and TNF-α). It is characterized by increased secretion of MMP-1, MMP-2, MMP-3, MMP-8, and MMP-13. Wittamer et al. first reported the association between Chemerin and inflammation. They observed a high concentration of synovial fluid (SF) in patients with arthritis [81]. Another study reported the association between high plasma Chemerin levels and the disease activity score for RA, and suggested that Chemerin may therefore be used as a biomarker of the disease [82]. Berg et al. described Chemerin and its receptor ChemR23, and revealed a significant increase in the levels of matrix metalloproteinases and proinflammatory cytokines after Chemerin stimulation. They reported that this receptor and its ligand may play a key role in joint inflammation and cartilage destruction [83]. In another study, after stimulation by IL-1β, the expression of Chemerin in chondrocytes increased. The interaction between inflammatory factors and Chemerin is also shown. Chemerin level is positively correlated with pain. Although our study shows that Chemerin level is negatively correlated with MMO, Chemerin may play a role as an inflammatory factor in TMJ disorder [84]. Chemerin levels gradually increase with the development of the disease. In addition, compared with patients without TMJ OA, the Chemerin level of TMJ OA patients is significantly higher, which supports that the protein may contribute to the development of TMJ OA [85].

### 3.4. TNFα Induced Chemotaxis and Chemokine Regulation in Synovial Fibroblast

TNF-α is one of the cytokines detected in the synovial fluid of patients with intercapsular lesions with disc displacement (DD) or OA [86]. TNF-α levels are also elevated in the synovium [87]. A study evaluating the role of temporomandibular joint synovial fibroblasts in chemokine release found significant increases in the expression of IL-8, GRO-α, MCP-1, and RANTES in the conditioned medium of TNF-α-treated synovial fibroblasts. The production of IL-8 and GRO-α increased soon after exposure to TNF-α (4 and 8 hrs), suggesting that neutrophil infiltration occurs in the early stages of the inflammatory response. The production of RANTES, a potent chemoattractant for CD4+/CD45RO+ memory helper T lymphocytes, increased less than other chemokines after 4 h of exposure to TNF-α. This suggests that T lymphocytes do not migrate to sites of inflammation as early as neutrophils and monocytes. These inflammatory cells produce inflammatory cytokines, such as TNF-α, matrix-degrading enzymes, and various oxidative metabolites. These enzymes and oxidative metabolism lead to extracellular matrix degradation. At the same time, inflammatory cytokines stimulate synovial fibroblasts to produce more chemokines (Figure 1). In conclusion, chemotaxis is a necessary factor for homeostasis, but the overproduction of chemokines appears to contribute to the destruction of joints [22].

### 3.5. Chemokines and RA

Inflammatory cells in TMJ OA are thought to play a pathological role in the development and persistence of inflammation by their ability to release degrading enzymes and various oxidative metabolites. Infiltrating cells recruited from the blood in synovitis are mediated by chemokines released by activated synovial cells. IL-8 is a member of a chemokine superfamily of low molecular weight cytokines. These have a chemotactic effect on neutrophil and T cell subsets as well as activating neutrophil function, which includes calcium mobilization, threshing, and respiratory outbreaks [36,88]. The infiltration of neutrophils and T cells into synovial tissue are thought to play a pathological role in the development and persistence of RA through their ability to release degrading enzymes and various oxidative metabolites [88,89]. It is also detected in synovial tissue in patients with RA. IL-1β is also elevated in the synovial fluid of RA patients, which is associated with joint pain and hyperalgesia. Furthermore, IL-1β plays a key role in amplifying and perpetuating inflammation and joint destruction [90,91]. Makiko et al. tested IL-8 in HTS cells treated or untreated with IL-1β with immunocytochemistry. All non-IL-1β treated HTS cells were stained negatively for IL-8. When treated with IL-1β, several cells in the cytoplasm around the nucleus were stained positive, while other cells were stained negative. These results suggest that HTS cells may have subpopulations with different functions, such as IL-1β reactive and unresponsive cells [92]. High levels of RANTES were detected in the body fluids of patients with RA and OA [93]. RANTES gene testing of synovial fibroblasts treated with or without IL-1β increased RANTES mRNA levels in IL-1β-treated synovial fibroblasts approximately 11-fold compared to untreated cells. When synovial fibroblasts were treated with 1 unit/mL of IL-1β, the generation of RANTES increase over the course of 48 h [75]. The production of MCP-1 is reported to increase when fibroblast-like synovial cells obtained from patients with TMD are stimulated by IL-1β [59]. MCP-1 production was significantly higher compared to co-cultures of IL-1β-treated TMJ synoviocyte-like cells (TMJ SC), and co-cultures of RAW264.7 cells with both cell-to-cell contact and without cell–cell contact. MCP-1 is strongly secreted in the inflamed synovium with inflammatory cytokine IL-1β-stimulated TMJ SCs, which further home monocytes and macrophages to the inflamed temporomandibular joint tissue. This results in increased adhesion of monocytes and macrophages to the synovium. IL-1β upregulated the expression level of ICAN-1 on inflamed synovial monocytes and macrophages (CD11a/CD18) of ICANSCs bound to TMJ SCs on TMJ SC, which indicates that IL-1β upregulates the adhesion ability of TMJ SCs to single cells and macrophages. Thus, cell-to-cell interactions between inflammatory cells and TMJ SCs, mediated by TMJ SCs and soluble factors on the surface of inflammatory cells (such as MCP-1 or adhesion molecules such as ICAN-1 and LFA-1), may exacerbate the symptoms of synovitis or facilitate its transition to chronic inflammation in TMJ (Table 1) [94]. 

## 4. Chemokine in Murine TMJ OA

Our previous study showed that β-catenin conditional activation mice, β-catenin (ex3)Agc1ER mice, and β-cateninAct mice exhibited a phenotype consistent with progressive TMJ disorder [95]. In addition, OA-like lesions appeared in the mouse TMJ cartilage in 9-week-old mice (3 weeks after the unilateral anterior crossbite (UAC) operation) [95]. To determine the molecular mechanism by which β-catenin regulates downstream target genes, we generated rat TMJ condylar chondrocytes cell lines [96]. We treated the cells with chemical compound, BIO, a GSK-3B inhibitor, and found that BIO significantly upregulated β-catenin protein levels in TMJ condylar chondrocytes (Appendix A). We then analyzed changes in the expression of multiple genes which have been reported to be involved in OA development. We found that the significantly upregulated gene by BIO is *Ccl2* in TMJ condylar chondrocytes (Figure 2F). We then examined the expression of other chemokines and their receptors, but did not find significant changes with BIO treatment. In this study, we also found that CCL2 protein levels were significantly upregulated in cartilage area from β-catenin activation mice and UAC-induced TMJ OA mice (Figure 2A). These novel data indicate that Chemokine signaling is possibly involved in TMJ OA disease through β-catenin signaling (unpublished), and further evaluation of chemokine is needed for therapeutic treatment in TMJ disease.

## 5. Conclusions and Perspectives

One of the possible causes of TMJ-related symptoms is arthritis. Among individuals with TMJ disorders, 11% have symptoms of Osteoarthritis (OA). It is estimated that approximately 15% of the world’s population suffers from OA, including TMJ OA. TMJ OA affects the cartilage, subchondral bone, synovial membrane, and other hard and soft tissues by causing changes like TMJ remodeling, articular cartilage abrasion, and deterioration. The TMJ OA is also involved in a sustained inflammatory process, and both metabolic and mechanical factors contribute to the early damage to the cartilage. Various chemokines and their receptors have been described in the arthritis model, and were targeted in different approaches including neutralizing antibodies, modified chemokines, and small-molecule receptor antagonists. Taken together, the available evidence suggests that specific inhibition of one chemokine, or one chemokine receptor, may have relevant clinical and biological effects in animal models. Very few studies have examined the role of the CCL2 chemokine in limb OA, but there are conflicting findings in their roles in cartilage. Although TMJ disorders are associated with symptoms, the causes are not well-proven. Since the etiopathogenesis of osteoarthritis is complex and is associated with multiple risk factors, the underlying molecular mechanism involved in TMJ OA development remains elusive. The treatment strategy for TMJ OA aims at preventing the progressive destruction of cartilage and the subchondral bone, which relieves joint pain and restores TMJ function. Therefore, our group will identify the role of chemokine in TMJ OA based on our preliminary data to address this critical knowledge gap. Several key discoveries in chemokine regulation of TMJ OA have laid a strong foundation to further investigate the role of specific chemokine signaling events in TMJ OA. Future studies should provide important insights into the mechanism of chemokine in TMJ OA, and define novel molecular targets for the development of effective therapies.

## Figures and Tables

**Figure 1 genes-14-00408-f001:**
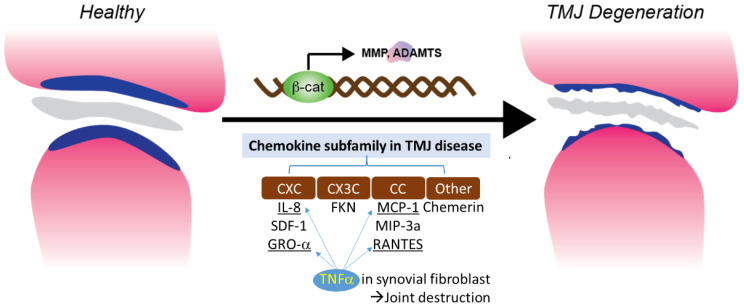
Chemokine subfamily associated with β-catenin and TNFα signaling in synovial fibroblast.

**Figure 2 genes-14-00408-f002:**
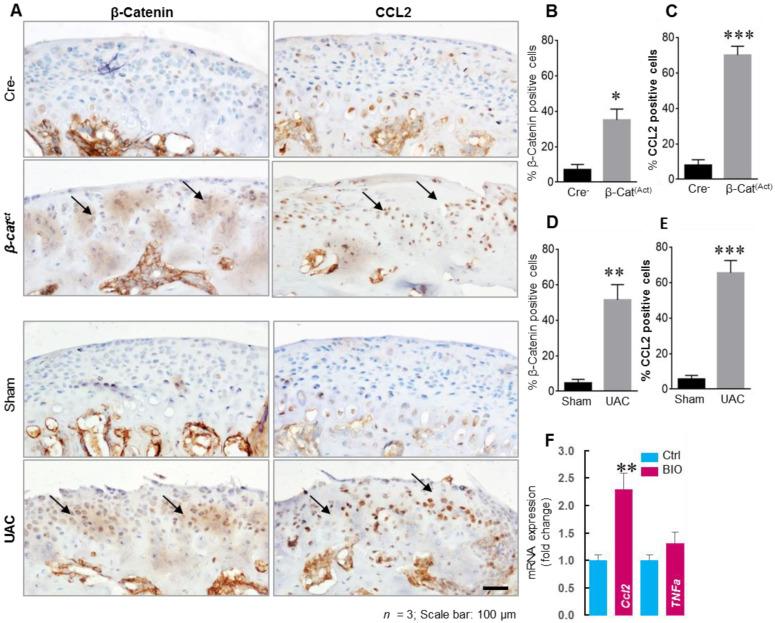
β-catenin and CCL2 expression increased in TMJ condylar fibrochondrocytes in TMJ OA mice and BIO upregulated *Ccl2* expression in rat condylar fibrochondrocytes. (**A**) TMJ tissues from β-catAct and Cre-mice were collected at the ages of 8-week-old for IHC analysis. The TMJ tissues of WT (C57BL/6) mice were also harvested 3 weeks after UAC operation for IHC analysis. Results showed that the levels of β-catenin and CCL2 increased in β-catAct mice compared to Cre-mice, and the same results were also shown in UAC-induced TMJ OA mice. Black arrows represent the typical positive expression cells in TMJ cartilage tissue. (**B**–**E**) Cells with positive staining for β-catenin and CCL2 were quantified (*n* = 3). Identically significant upregulation of CCL2 expression was also indicated in both β-catAct mice and UAC-induced TMJ OA mice. (**F**) Rat TMJ condylar fibrochondrocytes were treated with GSK-3β inhibitor, BIO, for 24 h. BIO (1 µM) significantly upregulated *Ccl2* expression in TMJ condylar fibrochondrocytes and increased β-catenin protein level (Appendix A). Statistical analysis was conducted using an unpaired *t* test. * *p* < 0.05, ** *p* < 0.01, *** *p* < 0.001.

**Table 1 genes-14-00408-t001:** Table summarizing the role of chemokine families in Temporomandibular Joint Disorder.

	Chemokines	Cells	Main Findings
CXC(α)	IL-8	macrophages, epithelial cells, airway smooth muscle cells, and endothelial cells	IL-8 is a powerful neutrophil attraction and activation cytokine in TMJ RA and OA. Upregulation of IL-8 in SMSCs caused by IL-1β also occurs by activating the NF-κB pathway
	SDF-1	hematopoietic stem cells	Activation of the SDF-1/CXCR4 signaling pathway regulates the expression of various inflammatory factors, including IL-1β, IL-6, TNF-α, and MMPs involved in TMJ pathology.
	GRO-α	neutrophils	Growth of new, small blood vessels in the TMJ synovium
CC(β)	MCP-1	monocytes, lymphocytes	IL-1β-stimulated temporomandibular joint synovial cells produce and release MCP-1, which is associated with the early stages of temporomandibular joint inflammation. MCP-1 may be a major factor in the onset, subsequent progression, and chronicity of TMJ synovial inflammation
	MIP-1α, 1β	monocytes, T lymphocytes	Recruiting inflammatory cells, wound healing, inhibition of stem cells, and maintaining effector immune response
	MIP-3α	lymphocytes and dendritic cells	Increase in MIP-3a may trigger the migration of dendritic cells, T cells, and B cells into the synovial tissue and body fluids of patients with TMJ-ID, and may lead to the onset and progression of inflammatory alterations in TMJ.
	RANTES	monocytes, T lymphocytes	RANTES/CCR1 signals are key signals that may play a synergistic role in GFP BMSCs for recruiting OA cartilage from the temporomandibular joint.
CX3C(δ)	Fkn (fractalkine, CX3CL1)	monocytes, natural killer cells, T cells, and smooth muscle cells	In the trigeminal nervous system, the persistent albumin-induced model of arthritis hyperphagia in TMJ activates the trigeminal tail subnuclear signal through the P2X7/CatS/FKN pathway
Other	Chemerin-ChemR23	dendritic cells, macrophages, adipocytes	The interaction of inflammatory factors and Chemerin increases the inflammatory effect. Chemerin levels were positively correlated with TMJ pain.

## Data Availability

Not applicable.

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
