# Peer review of "Chemokine Regulation in Temporomandibular Joint Disease: A Comprehensive Review"

_genes, 2023, doi:10.3390/genes14020408_

Round 1
Reviewer 1 Report
Many thanks for this submission. In the present paper the authors report their valuable evaluation regards to Chemokine regulation in Temporomandibular joint disease.
In the introduction section the authors should modify the phrase at Line 33 and report condromatosis as possible aetiology of temporomandibular joint disorder
Please modify as reported
“..Temporomandibular joint diseases (TMDs), including disc displacement (DD), condromatosis, and/or osteoarthritis (OA)…”
Please cite the following
Cascone P, Gennaro P, Gabriele G, Chisci G, Mitro V, De Caris F, Iannetti G. Temporomandibular synovial chondromatosis with numerous nodules. J Craniofac Surg. 2014 May;25(3):1114-5. doi: 10.1097/SCS.0000000000000667. PMID: 24739749.
Further conclusions should be shortened.
Author Response
Response: We appreciate the opportunity to consider additional information.
Please see line 42-44, we have modified as mentioned in reviewer’s point and added new references 8-9. In addition, we have deleted the text “The inflammatory process is coupled with the eventual degradation and abrasion of joint cartilage as well as the remodeling of the subchondral bone by the initiation of a local inflammatory response.’ to make shortened conclusion. Most essential parts are needed to describe our summary so we just deleted a few sentences.
Reviewer 2 Report
Hope that your future studies will give more answers for the mechanism of chemokine in TMJ OA and help the clinicians to find specific therapy for those kind of TMJ disease.
Author Response
Response: We are grateful for these encouraging comments. Yes, we are working hard to find more specific mechanism of chemokines in TMJ OA. It will be helpful to understand the pathological process of disease and develop therapeutic targets. Hope we could help the clinicians and researchers.
Reviewer 3 Report
Dear Author,
Chemokines play an important role in the pathogenesis of temporomandibular joint disease and knowledge on their role may have a key role for treatment of TMJ disease. In this review, authors discussed the correlation between chemokines (including MCP-1, MIP-1α, MIP-3a, RANTES, IL-8, 17 SDF-1, and fractalkine) and temporomandibular disorders.
This was a brilliant and very well organized review on this topic. However, the manuscript needs revision for language and grammar.
The manuscript is in line with the aim of the Journal, but there are some issues that should be added.
Title
Please report the study design. I suggest modifying the title to “Chemokine regulation in Temporomandibular joint disease: a comprehensive review” or similar.
Abstract
· Line13: “with a systematic review”. The first sentence is not clear. Please reformulate.
· Plase, add a comma before “and other structures”.
· The manuscript needs revision for language and grammar.
· Please add more Keywords.
Main text
I suggest improving the introduction section. Generally, the Introduction Section should report the definition of the “disease”, the epidemiological data, the signs and symptoms, the commonly used treatments. After, authors should report the recent literature on the topic, reporting the gap in the literature and the rationale of the study. Very few sentences on each point.
· Please refer to the Diagnostic Criteria for TMD (DC/TMD) Axis I. Thus, report that TMD could be divided in Group I: muscle disorders (including myofascial pain with and without mouth opening limitation); Group II: including disc displacement with or without reduction and mouth opening limitation; Group III: arthralgia, arthritis, and arthrosis.). (cite and refer to: Schiffman E, Ohrbach R, Truelove E, et al. Diagnostic Criteria for Temporomandibular Disorders (DC/TMD) for Clinical and Research Applications: recommendations of the International RDC/TMD Consortium Network* and Orofacial Pain Special Interest Group. J Oral Facial Pain Headache. 2014;28(1):6-27.). Moreover you cite as reference for RDC/TMD “John MT, Dworkin SF, Mancl LA. Reliability of clinical temporomandibular disorder diagnoses. Pain. 2005;118(1-2):61-69.” And not the correct one: “Dworkin SF, LeResche L. Research diagnostic criteria for temporomandibular disorders: Review criteria, examinations and specifications, critique. J Craniomandib Disord. 1992; 6:301–355.
· Moreover, improve epidemiological data, reporting that temporomandibular disorder is the second most common musculoskeletal disorder that causes pain and disability (Valesan LF, Da-Cas CD, Réus JC, Denardin ACS, Garanhani RR, Bonotto D, Januzzi E, de Souza BDM. Prevalence of temporomandibular joint disorders: a systematic review and meta-analysis. Clin Oral Investig. 2021 Feb;25(2):441-453. doi: 10.1007/s00784-020-03710-w. Epub 2021 Jan 6. PMID: 33409693. And Jin LJ, Lamster IB, Greenspan JS, Pitts NB, Scully C, Warnakulasuriya S. Global burden of oral diseases: emerging concepts, management and interplay with systemic health. Oral Dis 2016; 22(7):609-19. Doi: 10.1111/odi.12428).
· So, report the clinical manifestations as pain, decrease in the mouth opening, muscle or joint tenderness on palpation, limitation of mandibular movements, joint sounds and otologic complaints like tinnitus, vertigo or ear fullness, etcetera.
· Then, report the commonly used treatments for arthrogenous TMD (please cite and report Ferrillo et al. Efficacy of conservative approaches on pain relief in patients with temporomandibular joint disorders: a systematic review with network meta-analysis. Cranio. 2022 Sep 23:1-17. doi: 10.1080/08869634.2022.2126079.). As reported by this recent systematic review with meta-analysis, the most efficient treatments for arthrogenous TMD were occlusal splint and laser therapy.
· Please, better define the gap in the scientific literature and the rationale of the study.
· Please report the references according to the Instructions for authors.
Author Response
Response: We are grateful for the constructive recommendation to revise original manuscript. We made changes as follows
- Title: We have modified the title to “Chemokine regulation in Temporomandibular joint disease: a comprehensive review” as recommended.
- Abstract: Please see line 14 to 15. We reformulated with new sentences because we have mentioned the prevalence of TMJ in introduction part (line 35 to 36).
- Please see line 18. Corrected with “, and other structures”
- We have also checked our original manuscript by a native English-speaking colleagues to ensure the correct use of field-specific terminology in clear and accurate English.
- We have added more keywords. β-catenin; Unilateral anterior crossbite surgery
- We have reconstructed the original introduction followed by reviewer’s comments. New introduction included the definition of disease, the epidemiological data, the signs and symptoms and common treatments. We also mentioned the gap in the current literatures and the rationale study. Specific references were added. Please see the references, 1-17. We have also checked all the references in full text and corrected them.
Round 2
Reviewer 1 Report
Accept
Reviewer 3 Report
Authors modified the text according to the suggestions.
I found this work impactful and it fits well with in the scope of this journal.
In my opinion, it is suitable for publication.